# A Review of Potential Electrochemical Applications in Buildings for Energy Capture and Storage

**DOI:** 10.3390/mi14122203

**Published:** 2023-12-02

**Authors:** Jingshi Zhang, Rahman Azari, Ute Poerschke, Derek M. Hall

**Affiliations:** 1Department of Architecture, The Pennsylvania State University, State College, PA 16802, USA; razari@psu.edu (R.A.); uxp10@psu.edu (U.P.); 2Department of Mechanical Engineering, The Pennsylvania State University, State College, PA 16802, USA; dmh5373@psu.edu

**Keywords:** electrochemical energy harvesting, electrochemical energy storage, building skins

## Abstract

The integration of distributed renewable energy technologies (such as building-integrated photovoltaics (BIPV)) into buildings, especially in space-constrained urban areas, offers sustainable energy and helps offset fossil-fuel-related carbon emissions. However, the intermittent nature of these distributed renewable energy sources can negatively impact the larger power grids. Efficient onsite energy storage solutions capable of providing energy continuously can address this challenge. Traditional large-scale energy storage methods like pumped hydro and compressed air energy have limitations due to geography and the need for significant space to be economically viable. In contrast, electrochemical storage methods like batteries offer more space-efficient options, making them well suited for urban contexts. This literature review aims to explore potential substitutes for batteries in the context of solar energy. This review article presents insights and case studies on the integration of electrochemical energy harvesting and storage into buildings. The seamless integration can provide a space-efficient source of renewable energy for new buildings or existing structures that often have limited physical space for retrofitting. This work offers a comprehensive examination of existing research by reviewing the strengths and drawbacks of various technologies for electrochemical energy harvesting and storage, identifying those with the potential to integrate into building skins, and highlighting areas for future research and development.

## 1. Introduction

Industrialization and the associated fossil fuel combustion are responsible for significant greenhouse gas emissions and adverse impacts on the environment and its ecosystems. Examples of such impacts include rising sea levels, climate change, resource consumption, respiratory diseases, and biodiversity loss. Commercial and residential buildings are a key contributor to these impacts, accounting for 39% of energy use [1] and 35% of greenhouse gas emissions in 2021, which is a considerable share compared to the share of the other two sources of CO${}_{2}$ emissions, industry (28%) and transportation (37%) [2]. The reduction in building energy use and carbon emissions has therefore become a vital component of national and global policies to combat climate change.

Energy sources can be divided into two main categories: nonrenewable and renewable sources. Nonrenewable sources (petroleum, hydrocarbon, gas liquids, natural gas, coal, and nuclear energy) have taken millions of years to form becausethey have limited supply and are not quickly replaceable. In contrast, renewable sources like solar, geothermal, wind, biomass, and hydropower energy are naturally replenished and offer a sustainable alternative to nonrenewable sources [3].

Concerns about climate change and the unpredictability of fossil fuel availability are driving a shift toward the use of renewable energy sources in various sectors, including architecture. The widespread use of mechanical heating and air-conditioning systems in buildings has significantly increased energy consumption, emphasizing the necessity of transitioning to more environmentally sustainable alternatives [4]. As a result, architectural designers and engineers implement energy-saving strategies to reduce fossil-fuel-based energy consumption in buildings while adopting energy-generation technologies in buildings. Some of the energy-saving and decarbonization strategies in buildings include airtightness [5,6,7,8], increased insulation [9,10], high-performance windows, energy-efficient mechanical systems [11], low embodied carbon building materials [12,13], and developments in smart buildings [14]. The integration of energy-generating technologies into buildings will complement energy savings, adding an extra layer of efficiency [15,16]. As a result, there is an increasing number of studies that concentrate on generating energy locally for buildings.

Building skins are a crucial component of buildings that not only affect people’s impression of buildings but also play a critical role in maintaining a comfortable indoor environment. Architects and engineers have been developing high-performance energy-generating building envelopes for multiple decades now. BIPV serves as a noteworthy example of utilizing building skins as energy harvesting devices, gaining widespread acceptance and commercial availability. Implementing new technologies in buildings is often restricted by factors such as investment costs, lack of incentives, technological bottlenecks, and social acceptance [17,18,19,20]. These factors, along with low rates of energy conversion and the limited availability of inexpensive materials, have also slowed down the adoption of innovative renewable energy generation technologies for building skins. Funding and political demands have also affected the spread of new technologies. Nonetheless, the development of sustainable building envelope technologies is essential to reduce greenhouse gas emissions and promote a healthier environment.

Electrochemical technologies, such as batteries, fuel cells, and electrolyzers, have a broad range of applications and offer significant utility across various sectors. An example of an application is energy storage and backup power solutions for buildings [21,22,23,24]. Overall, these electrochemical technologies offer more than just a way to store energy for buildings. In their application, they act as a decentralized energy source; that is, they generate power right where needed through which they help stabilize the electricity grid. Additionally, they serve as adaptable storage systems that can adjust to a building’s specific energy needs, whether it is optimizing energy use during high-demand times or providing backup power during outages [25,26].

This paper provides a survey of building skin functions and reviews distributed energy generation and storage technologies available for buildings, with a focus on exploring and evaluating the potential of electrochemical technologies within building skin systems. It also critically analyzes the strengths and weaknesses of these electrochemical technologies and identifies areas for future research and development. Ultimately, the review aims to contribute to the advancement of sustainable building practices by identifying the most effective and efficient energy-generation and -storage solutions for buildings.

## 2. Building Skins and Functions

Building skins play a crucial role in providing occupants with a comfortable and healthy indoor environment while also minimizing energy consumption and reducing the building’s environmental impact. To achieve these goals, it is essential to consider various factors that contribute to occupants’ well being and energy efficiency [27]. Below are the main contributions of building skins.

Thermal Comfort Optimization: A well-insulated building skin is key to maintaining a comfortable indoor temperature throughout the year. By minimizing heat exchange with the external environment, a good insulation system helps to keep the interior cool during summer and warm during winter. Using high-performance insulation materials and design strategies, such as double-glazed windows, can significantly reduce heat transfer and improve energy efficiency [28,29,30].

Natural Ventilation and Daylighting: When designing a building, it is important to think about how windows and openings in the walls are placed and sized. This helps ensure that fresh air can flow in naturally and that there is enough sunlight coming in during the day. Having sufficient natural light not only makes the indoors more pleasant but also cuts down on the demand for electric lights. Smart and energy-efficient electrochromic windows serve as a strategy for controlling both lighting and heat transmission. Natural ventilation, both during the day and at night, is advantageous as it maintains fresh indoor air quality while also conserving energy. Consideration should be given to the size, placement, and control mechanisms of windows and other openings to optimize both ventilation and daylighting [31,32].

Moisture and Vapor Control: Incorporating water-resistance and vapor-resistance layers within the building skin is essential to prevent the negative impact of humidity on indoor spaces. By effectively managing moisture, building skins can protect the integrity of the structure, prevent mold growth, and maintain a healthy indoor environment. This can be achieved through the use of moisture barriers, vapor retarders, and proper sealing techniques to minimize water intrusion and control moisture levels [33,34].

Acoustic Performance: Building skins should also serve as effective acoustic barriers to minimize noise pollution from external sources such as traffic or construction activities. Incorporating sound-absorbing materials, acoustic insulation, and double-glazed windows with sound-reducing properties can help create a quieter indoor environment, enhancing occupant comfort and well-being. Careful attention should be given to the acoustic performance of building materials and the design of wall assemblies to minimize sound transmission [35,36].

Additional Aspects: Building skins can also contribute to reducing the building’s energy consumption and carbon footprint. For instance, incorporating renewable energy technologies such as BIPV can generate clean electricity, reducing the building’s dependence on grid electricity. Moreover, the use of sustainable materials in facade design can reduce the environmental impact of the building’s construction and operation. Green facades and green roofs can reduce the urban heat island effect, enhance air quality, and provide aesthetic benefits [37].

In summary, building skins should be designed with a comprehensive approach that considers thermal comfort, visual comfort, acoustic performance, and other aspects of indoor environmental quality. Optimizing all design aspects is impossible because they often contradict each other. For instance, windows provide natural lighting and ventilation, but they also represent the most vulnerable building component for energy loss. Adding other functions, such as photovoltaic panels, might change the properties of building skins. In such cases, compromising or finding a trade-off is essential. Sustainable building skins that incorporate energy-saving and energy-generation functions are preferred to play a role in addressing energy challenges and combating climate change [27].

## 3. Distributed Renewable Energy Technologies

The fact that residential and commercial buildings account for 39% of all U.S. primary energy use indicates the need for energy-efficient solutions in the built environment [1]. Using renewable energy is an alternative way to reduce the environmental impacts associated with fossil fuel use. Therefore, onsite energy generation through distributed energy resource (DER) technologies has gained popularity as a complementary energy source that supplements energy demands [38].

DER is a concept that aims to decentralize the traditional energy network and enables energy generation and storage onsite. By harvesting energy from renewable resources such as solar, wind, geothermal, hydropower, and biomass, buildings can offset their reliance on the grid system and reduce energy costs. According to the National Renewable Energy Laboratory (NREL), a system with an average size of 7.15 kilowatts, within a range of 3 to 11 kilowatts, is capable of fulfilling the energy needs of small residential buildings [39]. In a location with an annual average wind speed of 14 miles per hour (6.26 m per second), a 1.5-kilowatt wind turbine is sufficient to fulfill the energy requirements of a home consuming 300 kilowatt hours per month [40].

The grid system is not only centralized but also lacks flexibility, causing inconvenience in some areas due to geographical reasons. According to the International Energy Agency [41], 770 million people, mostly in Africa and Asia, live without a grid power supply. Decentralized solutions are the least costly way to supply power in these areas. Rural areas, such as off-grid islands, are not the only locations that can benefit from DER. Urban areas can also use onsite energy generation and storage as a backup or supplementary energy source to offset the burden of the grid [42].

DER technologies offer a promising solution to supplement energy demand in buildings, reduce our reliance on traditional energy sources, and promote sustainable development.

## 4. Building-Integrated Photovoltaic (BIPV) Technologies

Photovoltaic (PV) technology, which converts solar energy into electricity, is a good example of distributed renewable energy. By making use of the unique properties of semiconductors, solar radiation can be converted into a direct current through crystalline silicon [43].

Building-integrated solar panels have gained increasing popularity, as cities utilize solar energy to reduce the net load demand and promote onsite energy generation [44]. Today, solar energy accounts for over 5% of the electricity generated in the United States, which is nearly 11 times its share a decade ago [45]. The orientation of a photovoltaic (PV) installation significantly influences energy gain. An optimized PV installation incorporates a rotatable system to consistently capture maximal solar energy. Moreover, system design considerations such as PV sizing, energy storage, and the electrical and mechanical balance of systems play crucial roles in ensuring effective energy utilization [46].

The integration of PV technology can extend to different building elements, such as shading systems, rainscreen systems, curtain walls, double-skin facades, atria, and canopies [47]. PV systems in buildings can be designed as either grid-connected or stand-alone systems. Grid-connected systems can incorporate storage or operate without it, while stand-alone systems heavily rely on battery storage. Currently, batteries are the most common and commercially available technology for PV with electricity-storage systems in buildings, but future electrochemical technologies may also be employed [47].

The residential sector in the U.S. consumes 21% of the total energy (including end-use consumption and the energy losses within the electrical system related to retail electricity sales across various sectors), and the commercial sector consumes 18%, together making up the majority of energy consumption in all U.S. buildings [1]. However, with the increasing affordability of solar energy solutions and their adaptability to various climates, there is a promising opportunity for occupants to save on their energy costs in the future. By leveraging the potential of solar energy, occupants can not only contribute to a greener environment but also enjoy economic benefits by reducing their reliance on traditional energy sources [48,49]. The adoption of small-scale solar systems has been consistently rising, indicating a growing trend toward sustainability. According to the Independent Statistics and Analysis from the U.S. Energy Information Administration, by the year 2020, approximately 3.7% of single-family homes in the U.S. had already implemented such solar installations [50].

In this project, PV technology serves as the key component for collecting solar energy and converting it into electricity. Electrochemical components play a crucial role in storing energy to mitigate the intermittent problem of solar energy in buildings.

## 5. The Importance of Energy Storage Applications in Buildings

Generating electricity is a portion of the solution; storing it is equally important. Large electricity-producing plants typically still produce energy when is not needed. Distributed renewable energy, such as solar, is intermittent, which creates a challenge for the existing grid that requires a relatively stable load profile. Balancing high renewable energy penetration with grid stability requires a trade-off that cannot be avoided in today’s energy infrastructure. Energy storage is crucial to solving the intermittency problem [51,52].

Energy storage systems store the surplus electricity generated during periods of excess and release it when there is insufficient electricity available. There are three main ways of electrical energy storage: mechanical, chemical, or thermal systems. Most of the energy storage is done using mechanical methods, like pumped hydro, which makes up 95 percent of all storage. While chemical methods constitute a relatively small portion, there’s a noticeable trend of increasing battery storage each year. Over the past decade in the United States, the operational capacity has surged from 126.3 megawatts to 8827.1 megawatts [53].

Storage methods such as pumped hydro and compressed air storage facilities have certain limitations, such as low energy density, geographical constraints, and negative environmental impacts [54]. To address these issues, it is important to look into different ways of storage for urban areas where small-scale energy power devices are prevalent. One promising option is using electrochemical systems, offering several advantages, such as high energy density, flexibility, versatility, and high efficiency [54,55]. One notable advantage of electrochemical systems is their relatively fast response time, enabling them to react quickly whenever there is a demand for electricity [54]. This characteristic makes them highly suitable for urban environments where getting power fast is crucial.

While certain electrochemical systems might come with expenses, it is important to understand that they provide a broader selection of storage choices for buildings [51]. This variety allows for customized solutions that can match the special needs of the built environment.

## 6. Electrochemical Applications in Buildings

Electrochemistry is a discipline that studies the relationship between electrical energy and chemical change. In plain language, it can be understood as an energy conversion between electrical energy and chemical energy [56]. For instance, hydrogen and oxygen can react and produce electricity through electrochemical energy conversion devices known as fuel cells. Another common example is the battery that can provide electricity from stored chemical energy. Some fundamentals are needed to understand an electrochemical device’s behavior and performance, such as charge transport, electrochemical thermodynamics, and electrochemical kinetics. When these electrochemical devices are integrated into buildings, they have the potential to alter the characteristics of building skins, impacting factors such as thermal performance and moisture control. Vice versa, electrochemical devices integrated into buildings always deal with real-world operating conditions, which may be different from ideal conditions.

The electricity storage patches up the mismatch between load demand and renewable energy resources generation profile [57]. Lund et al. quantified the mismatch of renewable sources at the building level. This mismatch is attributed to hourly variations in energy production and consumption within buildings. The findings indicate a 63% mismatch for PV and a 39% mismatch for wind [58]. Electrochemical energy storage is versatile. Not only being used for grid-scale energy storage and automobiles but electrochemical applications such as batteries have been already used in buildings to support intermittent renewable energy [57]. Nowadays, with the emergence of many electrochemical technologies, how to choose a suitable energy storage method for buildings is a very important topic. Energy storage density plays a critical role in determining the storage capacity of a system [59], whereas power density is focused on how quickly the energy can be delivered. The measure of energy density can take the form of volumetric energy density, commonly expressed in units such as watt-hours per liter (Wh/L). This metric signifies the amount of energy that can be stored within a specific volume. Gravimetric energy density, on the other hand, pertains to the quantity of energy that can be stored in a given mass and is typically quantified in units like watt-hours per kilogram (Wh/kg). Volumetric power density is often assessed in units like watts per cubic meter (W/m${}^{3}$) or watts per square meter (W/m${}^{2}$), denoting the rate at which power can be delivered in relation to volume or area, respectively. When it comes to buildings, it is essential to have flexible energy storage capacity without a substantial increase in costs. For example, it is meaningful to figure out how much space inside buildings is needed to achieve a particular storage capacity and to identify the maximum storage capacity possible without making significant changes to the building structure. Additionally, considering power density is crucial, as buildings may require electricity promptly, although not necessarily as rapidly as vehicles do. Right now, batteries’ energy and power density performance is in between fuel cells and capacitors [60]. They are relatively reliable and technically mature. However, the widespread implementation of batteries does not mean other technologies are not possible in the future. Many researchers have been working on fuel cells and supercapacitor applications in built environments in cities. When implementing energy storage technology in buildings, both gravimetric energy density and volumetric energy density are important. Designers weigh trade-offs when creating architectural plans. Volumetric energy density becomes crucial when the storage system occupies specific spaces within the building, such as the basement, where higher volumetric energy density allows for space-saving storage solutions. On the other hand, gravimetric energy density will be a factor in situations where the buildings require lighter components for efficient energy storage.

When reviewing electrochemical devices, several parameters need to be considered. The common parameters include energy storage and power density capacities, response time, operational conditions (temperature), round-trip efficiency, and lifetime [61]. However, if these devices are intended for building applications, additional special considerations come into play. Safety becomes a major concern since buildings are designed for occupancy activities. Factors such as fire safety, strength safety, and overall safety measures should be prioritized. Durability is also crucial for building skins, as they are often subjected to various weather conditions. Therefore, materials chosen for electrochemical devices in building skins should be weather-resistant, water-resistant, and fire-resistant where possible. Efficiency, cost, and maintenance are other practical aspects that should be taken into account for building applications [62].

Research on electrochemical energy storage methods, including batteries, reversible fuel cells, and supercapacitors, has gained considerable attention in building applications. Among these methods, batteries currently dominate the field, particularly when paired with renewable energy sources like solar or wind power. For instance, lithium-ion batteries have been extensively studied due to their favorable combination of gravimetric and volumetric energy densities [63,64,65]. Reversible fuel cells present an alternative option and have been compared to batteries in terms of their levelized cost of storage for buildings [66,67]. These fuel cells are also utilized as cogeneration systems in buildings, offering additional benefits [68,69]. Supercapacitors, on the other hand, have been integrated into building components and materials to enhance energy efficiency [24,70,71]. It is important to note that they work just as well as batteries, but right now, they might not be affordable for some building projects [72]. As technology gets better and cheaper, these different ways of storing energy might become more popular for buildings. The upcoming sections explain foundational concepts and illustrate them with contemporary examples. These examples showcase recently completed projects, with some drawn from the latest literature. The criteria for selection are rooted in the following considerations: 1. Pertinence to architecture or communities; 2. Recent research discoveries or actual projects.

### 6.1. Traditional Batteries and Flow Batteries

Batteries have been categorized into two groups. Primary batteries are batteries that cannot be recharged. Common examples are zinc–manganese dioxide alkaline cells and lithium metal batteries [73]. Secondary batteries, which are rechargeable, are mainly discussed in this review. Common examples of secondary batteries are lead acid batteries and lithium-ion batteries [74]. Batteries are usually designed as sealed systems that require less operation and maintenance, but this design also causes other problems, such as degradation and self-discharging [75,76]. Lithium-ion batteries stand out due to their extended life cycle and high energy density, making them a favorable option for renewable energy storage [77]. Their suitability for Building Integrated Photovoltaic (BIPV) systems is particularly promising. Tervo et al. [64] demonstrated that their lithium-ion system competes effectively with the grid in terms of long-term costs. Their model reveals a cost of USD 0.11 per kilowatt-hour (kWh) for the lithium-ion system, whereas the average electricity cost in California, for instance, is approximately USD 0.3 per kWh.

Though less common, redox flow batteries are another interesting option for building applications. Redox flow batteries differ from traditional batteries by using two external electrolyte tanks, which are circulated using pumps to facilitate redox and oxidation reactions for energy storage and release [78]. As these are also secondary batteries, redox flow batteries can be recharged. For building applications, this design offers several advantages over other batteries. Firstly, they are scalable by adding more electrolyte solutions without affecting power output. Additionally, they have long lifespans due to the absence of permanent self-discharge, as reported in the study by Weber et al. [79]. Furthermore, flow batteries are safer than traditional batteries since they use non-flammable and less toxic electrolyte solutions [80]. Their efficiency and response times are also noteworthy, as flow batteries can quickly react to provide electricity. These advantages, combined with its distinctive design and traits, make flow batteries a promising technology for storing energy within buildings.

Despite numerous studies demonstrating the high efficiency of redox flow batteries (RFBs), their widespread application in the energy and building sectors is still limited. There are some examples of RFB usage in grid-scaled infrastructure, such as the world’s largest RFB power station—Dalian Constant Current Energy Storage Power Station Co., Ltd. [81], in Dalian, China. Other projects in the USA are undertaken by Largo Clean Energy and partners in Wilmington, Massachusetts, and Quino Energy, Inc. in Menlo Park, CA, USA. These initiatives, supported by the DOE, aim to reduce the cost of energy storage by 90% within 10 years [82]. There are residential vanadium flow battery applications from companies like StorEn based in South Carolina [83] and instances of flow batteries being installed in communities as micro-grid power plants, such as the one in San Diego that powers 1000 homes. These projects emphasize the long lifespan and cost-saving benefits of redox flow batteries in the long run.

The integration of redox flow batteries into buildings has not been extensively studied, although there are some theoretical analyses available at the pack level. Nguyen group [84,85] analyzed the performance of a photovoltaic–vanadium redox battery microgrid system under various loads and temperatures. They provided detailed studies on the microgrid system and the VRB schematic diagram, which could be applicable on a building scale. Parameters such as solar insolation, temperature, voltage, and current were monitored and recorded at 5-second intervals. State of charge, open-circuit voltage, discharge, and charge performance were analyzed. Two operational modes were designed for performance studies: “renewable mode”, where a small building was powered by the VRB and PV, and “grid mode”, where the building was powered by the grid while the VRB was charged using PV. A schedule was created to control the modes to ensure the building was fully powered by the grid, VRB, or PV. In their 2015 study, the same group, led by Qiu, optimized the size of the PV-VRB system to maximize efficiency. Qiu et al. [86] conducted another study on a standalone system, where the PV supplied energy during the day and excess energy was converted to VFB, allowing the building to be powered solely by the VRB during the night. However, the specific location of the energy-storage devices in the building was not explicitly mentioned. It was noted that the temperature of these storage devices was regulated by an HVAC system, suggesting their placement within the building itself. Zhang et al. [87] proposed a computational method to assess the cost and efficiency of PV-RFB systems in residential buildings, addressing concerns regarding installation costs. The simulation indicated a payback period of 3–5 years for the system. Other hybrid systems have also been developed, such as a multi-story building system comprising PV, hybrid gravity power, and vanadium redox flow batteries [88]. These hybrid systems housed in walls and basements cater to the needs of more complex and demanding buildings.

### 6.2. Supercapacitors

Compared to fuel cells and flow batteries, supercapacitors have high power density, which means the charging and discharging speed is higher than that of reversible fuel cells and flow batteries. However, the energy density is limited by the special structure of supercapacitors [89]. The capacitance of supercapacitors is positively proportional to the surface area and negatively proportional to the distance between the double layers. The capacitance is also related to the electric constant and dielectric constant, which depend on the property of the material between the plates [90]. Supercapacitors achieve energy storage by utilizing the electric double layer (EDL). The theory of the EDL was originally proposed by Hermann von Helmholtz in 1853. Further developed supercapacitors were first commercially available in 1971, and the low internal resistance supercapacitor was invented in 1982 for military purposes [91].

Supercapacitor applications in buildings are relatively new, with studies emerging over the past five years. The Fraunhofer Institute for Solar Energy Systems (ISE) has undertaken projects that focus on integrating solar cells with supercapacitors at the device level, resulting in a device known as a photo-supercapacitor. One group, led by Delgado Andres [92], introduced a novel solar charging system. Their device features a three-electrode configuration comprising high-performance organic solar cells (OSCs) combined with nitrogen-doped carbon supercapacitors. The highest energy density of 1.6 × 10${}^{-4}$ Wh cm${}^{-2}$ was achieved at 0.2 mA cm${}^{-2}$. Although the energy conversion efficiency of this system tested at 2% is lower than the 17% efficiency achieved by a standard approach from the literature, it represents a new device capable of simultaneous energy harvesting and storage. Another group, Berestock et al. [93], also from the Fraunhofer ISE, pursued the concept of combining PV cells with capacitors using a different technology at the device level as well. They integrated a perovskite solar cell with a mesoporous carbon capacitor, achieving a peak energy conversion efficiency of 11.5% with an energy density of 4.27 μ Wh cm${}^{-2}$. These two studies share a similar design for photo-supercapacitor devices: the PV cell on top harvests solar energy, while the supercapacitor beneath it stores the converted energy as chemical components. The overall efficiency of the system is defined as the product of conversion efficiency and storage efficiency. It is worth noting that both studies face limitations, such as the high cost of supercapacitors and small-scale testing without application to an entire building. Both innovative devices hold the potential for utilization similar to PV panels within buildings. They can be installed on rooftops or integrated into building envelopes, enabling them to effectively harvest and store energy.

Another research publication [94] demonstrated the integration of 3D-printed electrochemical devices with bricks for energy storage, also at the device level. They developed a novel brick design by incorporating printed supercapacitors into the voids of brick insulation. The use of a Ti3C2@PPy-coated electrode exhibited favorable conductive and capacitive performance. The most favorable energy density outcomes are achieved at 2.64 Wh kg${}^{-1}$. If this “smart brick” can seamlessly integrate with the entire building system, it could present a promising and novel option for energy storage.

While these studies showcase promising developments in integrating supercapacitors into building applications, challenges such as cost and scalability remain to be addressed. Nonetheless, these innovations highlight the potential for utilizing supercapacitors for energy storage in building skins.

### 6.3. Fuel Cells

Fuel cells are devices that convert the chemical energy of a fuel and an oxidant into electrical energy [95]. The key components of a fuel cell are an ion conductor, electrodes, and a catalyst. The anode and cathode are the two electrodes that facilitate two different reactions—oxidation and reduction. Oxidation generates electrons, while reduction accepts electrons [96]. The potential difference between these two electrodes creates an electric current. The ion conductor is the component that conducts ions between the two electrodes [89]. The catalyst is usually added to the cell to accelerate the reaction [97]. In many cases, gas-diffusion layers are also necessary to diffuse gas to the key components [98]. Typically, a fuel cell is not a single “sandwich” structure. Like batteries, many fuel cells are stacked together to produce higher energy and power [96].

DOE describes six common types of fuel cells, namely polymer electrolyte membrane fuel cells (PEMFCs), alkaline fuel cells (AFCs), phosphoric acid fuel cells (PAFCs), molten carbonate fuel cells (MCFCs), solid oxide fuel cells (SOFCs), and direct methanol fuel cells (DMFCs) [97,99] (Table 1). Each of them works in different conditions but has similar energy efficiencies ranging from 40% to 60% [100]. In addition to the six above-mentioned types, a microbial fuel cell (MFC) is another type that is relatively uncommon.

#### 6.3.1. Solid Oxide Fuel Cells (SOFCs)

Solid oxide fuel cells (SOFCs) use non-porous ceramics as ion conductors. These cells can achieve an impressive efficiency of 80–85 % if a waste heat recycling system is in place [101]. However, SOFCs require high-temperature operational conditions, usually around 900 °C [102]. One major advantage of these cells is that they are tolerant to sulfur and carbon monoxide, allowing them to use some fossil fuels directly from coal as an energy source [97,103]. The ion conducted through an ion-conductive ceramic is O${}^{2-}$. The cathode typically uses La${}_{0.8}$Sr${}_{0.2}$MnO${}_{3}$ (LSM) and the anode is mixed with Ni-YSZ [104]. Solid oxide fuel cells can be regenerative cells like PEM regenerative fuel cells. Therefore, the same device can be used for either galvanic or electrolytic purposes [105]. SOFC is well developed and has a low operating cost, but the start-up time is longer because of the high operating temperature [97,99].

One key advantage of SOFCs is their high efficiency, making them a potential cogeneration system for buildings [106]. Several studies have explored the use of SOFCs in buildings at the pack level, including work by [100,107,108]. Ref. [109]’s computational model investigated the parameters that influence SOFC performance and simulated different system configurations for small- and large-scale buildings in various weather conditions. The results showed that efficiency depends on system configurations, building types, and weather conditions. Meanwhile, Ref. [108] demonstrated that SOFCs can be integrated with cooling and heating systems in buildings, reaching an impressive energy efficiency of 60% in Aspen Plus. These research studies demonstrate that SOFC systems can generate electricity, and buildings can make use of the heat produced as a byproduct of SOFCs. When integrated with cooling and heating systems in building designs, they can improve overall energy efficiency, but there is no study that indicates SOFCs can be used as building skin components.

#### 6.3.2. Alkaline Fuel Cells (AFCs)

Alkaline fuel cells display cost effectiveness and quick reaction times [110]. The electrolyte in it is made up of a solution containing potassium hydroxide (KOH), and this can be sensitive to CO${}_{2}$ in the air, leading to potential maintenance problems [108]. The operational temperature can be below 100 °C [97,111]. A range of catalysts, including nickel (Ni) or silver (Ag), can be used, and OH- is conducted through electrolytes [112].

The recent launch of a micro-combined heat and power (CHP) alkaline fuel cell prototype by PWWR Alkaline Fuel Cell Power Corp named Jupiter 1.0 offers a new possibility for small buildings to generate electricity with a high efficiency of 90% [102] at the pack level. Hydrogen is converted into both heat and electricity. This device is potentially stored in buildings or onsite to serve as a backup energy production system. Behling [103] developed a simulation model to evaluate pack-level AFC-based CHP systems and compared them with other CHP technologies such as gas engines, Stirling engines, PEM, or SOFC-based fuel cell systems, in terms of their electricity efficiency defined as HHV and thermal efficiency. While the AFC-based CHP system showed similar total efficiency as the other technologies, its thermal efficiency was not as good. Since a building skin component will be created in the upcoming study, its sensitivity to CO${}_{2}$ [113] makes it less than ideal for use in environments exposed to unavoidable contamination.

#### 6.3.3. Phosphoric Acid Fuel Cells (PAFCs)

The phosphoric acid fuel cell (PAFC) is named after its electrolyte, which consists of phosphoric acid soaked in a matrix [114]. It can be operated at temperatures ranging from 150 °C to 200 °C [115,116,117]. H+ ions are conducted through electrolytes, and concentrated phosphoric acid offers relatively stable thermal and electrochemical performance. However, the use of platinum as a catalyst increases the cost of PAFCs [108].

PAFCs were the first fuel cell technology to be welcomed into residential and commercial buildings, with UTC Power being one of the biggest North American companies producing PAFC products. Their stationary fuel cells have been successfully proven to provide electricity to buildings such as hotels, and educational institutions with lower energy costs and high system efficiency [118]. While PAFCs have several advantages, including their ability to operate at relatively low temperatures with a range of 150 °C to 200 °C and their tolerance of CO${}_{2}$, the use of platinum as a catalyst increases their cost [115]. Ruan et al. [119] assessed onsite stationary cogeneration systems by comparing gas turbines, gas engines, and PAFC with heating and cooling capabilities. Despite the PAFC system having a longer payback period, the research demonstrated that it offers greater energy savings in specific building types, such as hospitals. PAFCs can be used as building energy cogeneration systems but are not ideal for flexible building skin-cladding components.

#### 6.3.4. Molten Carbonate Fuel Cells (MCFCs)

Molten carbonate fuel cells require a high operating temperature, usually around 600 °C to 700 °C [120]. In MCFC, CO_3_^2−^ is the ion conducted in the electrolyte [121]. Solutions of lithium, sodium, or potassium carbonates soaked in the matrix are common electrolytes [97].

Santa Rita Jail made use of MCFC as an additional power source to the grid and solar panels. As mentioned in their energy report, the energy from MCFC accounted for 50% of total electricity usage. Heat as the byproduct of chemical reaction could provide 18% of heating for the jail. It is a stationary power plant located onsite [122]. The relatively high operating temperature makes it not ideal for building skins.

#### 6.3.5. Direct Methanol Fuel Cells (DMFCs)

Many fuel cells primarily rely on hydrogen as their fuel source. However, direct methanol fuel cells offer the advantage of using readily available methanol as their fuel [123]. They can operate at temperatures ranging from 60 °C to 100 °C [97]. Nonetheless, one drawback of this fuel cell type is the production of CO_2_ as a byproduct, which is something we aim to minimize or avoid [123].

#### 6.3.6. Proton Exchange Membrane Fuel Cells (PEMFCs)

The proton exchange membrane fuel cell (PEMFC), also known as the polymer electrolyte membrane fuel cell, utilizes a perfluoro sulfonated acid polymer layer as the electrolyte, which can only conduct protons (H+) [124]. This cell typically employs platinum as the catalyst, allowing it to operate at a relatively low temperature (below 80 °C), making it easier to integrate with building skin components [125].

Regenerative fuel cells can be either discrete or unitized. The unitized regenerative fuel cell (RFC) is receiving more attention due to its multifunctional properties [126]. However, one of the most challenging problems in making a highly efficient reversible PEMFC is ensuring that the oxygen-side catalyst layer works effectively in both the galvanic and electrolysis modes while also providing anti-corrosion and long-lasting cells. Further research is needed to address this challenge [127].

PEM fuel cells have been studied as part of the cogeneration system of buildings at the pack level. In a study by Ashari et al. [128], a PEM fuel cell system was designed to cover electrical, hot water, heating, and cooling loads in a residential building. They estimated the residential building’s load and then designed eight stacks of fuel cells with 8.4 kW power capable of providing enough energy for the building. Natural gas was used as fuel, and the electricity cost was 5.41 USD/kWh. Similar studies, such as those conducted by Ham et al. [129] and Chahartaghi et al. [130], have also developed energy models for building cogeneration applications. The efficiency of these systems ranges from 45% to 82%, which is optimistic for further development.

Chadly et al. [63] conducted a cost simulation for energy storage systems using PV as the original renewable energy source. The electricity generated by the PV system was stored in Li-ion batteries, reversible SOFCs, or PEM reversible fuel cells. Although the cost of PEM-based RFC (39.17 USD/kWh) is not competitive with the other two options (5.49 USD/kWh for Li ions, 26.45 USD/kWh for SOFCs), it has other merits such as low operating temperature and totally clean byproducts, which make it a potential player in the future building skin market.

#### 6.3.7. Microbial Fuel Cells (MFCs)

Microbial fuel cells (MFCs) are a promising technology because they use microorganisms as their fuel source and can potentially generate electricity from a variety of organic materials [131]. They are named for their fuel, not their electrolyte, and typically use a proton-exchange membrane to conduct H+ ions [132]. It is worth noting that microbial electrolysis cells use a different design and system and as a result are not regenerative like PEMFC [133].

In 2019, You et al. [134] published a device-level study on the use of microbial fuel cells (MFCs) in buildings. The study validated the feasibility of converting buildings into micro-power stations by using MFCs. The idea is that bricks could be used as MFC reactors, with microorganisms generating electricity through chemical reactions. The researchers tested two common bricks used in the UK, as well as one handmade brick, and designed each brick differently. The MFC bricks were tested in ambient conditions and fed with either municipal wastewater or human urine. The output of the MFC bricks was recorded in electrical potential (volts), and the experimental results indicate that MFCs can be integrated with commercially available building bricks, with a maximum power output of 1.2 mW per brick. While the output was relatively lower than existing literature due to the non-optimized configuration and the use of cheap, low-efficiency catalysts, it is still considered to be a promising alternative for onsite power generation. Further studies of such fuel cell applications are worth exploring.

**Table 1 micromachines-14-02203-t001:** Different types of fuel cells.

Fuel Cell Type	Common Electrolyte	Operational Temperature	Stack Size	Electrical Efficiency (LHV)	Advantage	Disadvantage
SOFC	Y${}_{2}$O${}_{3}$-stabilized ZrO${}_{2}$ [104]	600–1000 °C [101]	1 kW–2 MW [111]	80–85% [101]	Electrolyte is solid; reuses heat waste; fast kinetics [97,99,103]	High operating temperature [102]
AFC	KOH retained in matrix [108]	<100 °C [97,111]	1–100 kW [111]	60–90% [102,111]	Flexible to use a wide range of catalysts; cost-friendly; quick reaction [110,112]	Very sensitive to CO_2_ [113]
PAFC	Liquid phosphoric acid in SiC [114]	150–200 °C [115,116,117]	5–400 kW [111]	40%	Tolerant to CO${}_{2}$, heat waste can be recycled [111]	Expensive catalyst [108]
MCFC	Molten carbonate in LiAlO${}_{2}$ [97]	600–700 °C [120]	300 kW–3 MW [111]	50% [122]	Cheap catalyst; reuses heat waste [111]	High operating temperature causes damage to materials [120]
DMFC	Polymer electrolyte membranes [135]	60–100 °C [97]	10 MW [136]	40–90% [137]	Uses methanol as fuel; low operational temperature	Produce greenhouse gas [123]
PEMFC	Polymer electrolyte membrane	<80 °C [125]	<1–100 kW [111]	45–82% [130]	Solid electrolyte; low operational temperature; rapid start-up [124,125]	Difficult thermal management and water management; sensitive to contaminants [111]
MFC	Proton exchange membrane [132]	4–55 °C [130,138,139]	258 W m${}^{-3}$ (Power output) [140]		Microorganisms as fuel [131]	Very low efficiency

### 6.4. Two Electrochemical Energy Storage Applications for Building Skins in This Research

For the majority of electrochemical applications discussed earlier, it is evident that they are primarily employed as stationary power plants rather than for energy-storage purposes (Table 2). In simpler terms, some applications only serve as co-generation systems to supplement grid electricity. Their potential as contributors to distributed energy resources is significant, particularly when they utilize clean energy sources. As mentioned previously, our goal is to create a building skin component capable of harvesting renewable energy and storing and regenerating energy. Hence, a reversible system that can store energy is preferred for future design and development.

When thinking about using these technologies in building skins, there are additional aspects to be considered. In this research, a redox flow battery (RFB) and reversible proton exchange membrane fuel cell (RPEMFC) have been chosen as two options for integration into building skins after looking at different electrochemical technologies. The reasons are as follows. 1. Scalability: Energy storage options like lithium-ion batteries and supercapacitors cannot scale up as well as flow batteries and fuel cells. 2. Operational conditions: We looked at different types of fuel cells and their operational conditions, like operational temperature and how well they handle CO${}_{2}$. We prefer fuel cells with lower requirements in these areas and therefore ruled out options like PAFC, MCFC, SOFC, and AFC, which have higher demands. 3. Carbon neutralization: DMFC produces CO${}_{2}$ as a byproduct, which is not sustainable.

**Table 2 micromachines-14-02203-t002:** Examples or studies of electrochemical devices for buildings.

Electrochemical Device Type	Subtype	Examples or Studies	Special Features	Advantages	Disadvantages
Batteries	Lithium-ion batteries	[141]	Round trip efficiency 82–89% (pack level)	Long life cycle; high energy density [77]	Degradation; Safety concern [142]
	Redox flow batteries	PV-RFB (pack level) [84,85,86]	The system attained 80% advertised efficiency	High-efficiency energy storage system for microgrid	Operating cost still relatively high
	Redox flow batteries	PV-Gravity storage and RFB (pack level) [88]	Energy supply from renewable resource is 47.77%		
Supercapacitors		Organic solar cell-supercapacitor (device level) [92]	Highest energy density 1.6 × 10${}^{-4}$ Wh cm${}^{-2}$	Harvesting and storing energy within a single device	Low energy conversion efficiency of 2%
		Perovskite solar cell-supercapacitor (device level) [93]	Energy density 4.27 μ Wh cm${}^{-2}$	Harvesting and storing energy within a single device	Low energy conversion efficiency
		3-D-printed supercapacitor bricks (device level) [94]	Energy density 2.64 Wh kg${}^{-1}$	Good cycle life: Maintained 81.4% of the initial value following 6000 charge/discharge cycles	
Fuel cells	Alkaline	PWWR Jupiter 1.0 (pack level) [102]	Efficiency of 90%	High efficiency and quick reaction	Sensitive to CO${}_{2}$
	Phosphoric acid	UTC power (pack level) [118]	40–42% electrical efficiency	Relatively low operating temperature	
	Molten carbonate	Santa-Rita Jail (pack level) [122]	Heat as byproduct provided 18% heating for the building		Slow startup
	Proton exchange membrane	PEM combined heat and power system (pack level) [128]	Efficiency of 45–82%; used as cogeneration system of buildings	Low operating temperature	Expensive
	Microbial	Microbial fuel cell bricks (device level) [134]	Used waste water and urine to generate electricity with a power of 1.2 mW	Integrated with bricks	Low efficiency

Both RPEMFCs and RFBs possess a comparable assembly structure (Table 3), and our goal is to explore their individual strengths and weaknesses when they are employed in the integration of building skins.

The subsequent phase of this research involves proposing and evaluating new building skin solutions that function as decentralized energy sources. This is achieved through BIPV and electrochemical energy storage technologies, notably reversible PEM fuel cells and redox flow batteries. The advantages of incorporating these devices into building skins encompass the facilitation of multifunctional building skins and streamlined installation of energy components. These advantages are especially useful when renovating existing buildings, as they eliminate the need for extra space to fit energy systems and offer a more space-efficient option for building designs.

Redox flow batteries offer extended life cycles and cost–benefit considerations for long-term applications. Moreover, they are relatively safer due to the absence of flammable materials and can be conveniently scaled up. Given these advantages of flow batteries, exploring their implementation in building skins holds significant promise [143].

Reversible PEM fuel cells allow for increased energy-storage potential by expanding the size of the storage system, like the way RFBs store energy. RPEMFCs are promising energy storage technology that can support the grid [144]. Hydrogen is the stored chemical substance, which is a clean source of energy with zero CO${}_{2}$ emissions [145]. The U.S. Department of Energy is encouraging the use of hydrogen in various applications [146]. Out of the six types of fuel cells, considering their reversibility and operational temperature, reversible PEM fuel cells appear to be the most suitable choice for building skins.

**Table 3 micromachines-14-02203-t003:** Comparison of two chosen electrochemical devices for study.

	Reversible PEM Fuel Cells	Redox Flow Batteries
Reactants	Gas	Liquid
Operational temperature	<120 °C [91]	−20 °C to 50 °C [147]
Toxicity	No toxic gas emission	Sometimes include toxic materials
Flammability	Yes	No
Components needed	Power cells; water and gas tubes; hydrogen containers; wires; pumps	Power cells; tubes; electrolyte solution containers; wires; pumps
Roundtrip efficiency	50% [148]	70–80% [149]
Life cycle and life time	>5000 cycles [148]	1500–15,000 cycles [149]
Energy density (Wh/kg)	400 Wh/kg [150]	10–35 Wh/kg [151,152]
Power density	10–500 W/kg [153]	100–166 W/kg [152]

## 7. Fuel Cells and Fuel Storage Safety Concerns

When integrating fuel cells into building skins, a significant challenge arises in the form of hydrogen storage. Hydrogen, a flammable gas, poses safety concerns and presents difficulties in containment.

To deal with this problem, the National Fire Protection Association (NFPA) 853 standard [154] gives instructions for the installation of stationary fuel cell power in buildings. This rule helps make sure fires are less likely to happen and easier to control. This standard serves to enhance fire prevention and protection. It provides clarity on various aspects, including the placement, interconnection, supply, storage arrangement, ventilation, exhaust, and fire safety of fuel cell power systems.

An important consideration is that the storage location for the equipment must be segregated from other sections of the building by a fire barrier wall with a minimum of one hour of fire resistance. Aligning with such requirements, our research considers exterior walls with fire-resistant properties in compliance with relevant codes. Additionally, given the outdoor nature of the equipment, protection against weather elements such as rain, snow, ice, and lightning is imperative.

Numerous other standards and codes govern different facets of the system’s implementation. These include the NFPA 2 hydrogen technologies code [155], NFPA 855 Standard for the Installation of Stationary Energy Storage Systems [156], ANSI/CSA FC1-2014 (IEC 62282-3-100:2012, MOD) (formerly ANSI Z21.83) [157] for Fuel Cell Power Systems, and the 2021 International Fire Code (IFC) [158] Chapter 6, which pertains to Building Services and Systems.

## 8. BIPV and Electrochemical Storage Applications Opportunities and Constraints

Based on the investigations presented so far, our team has developed two types of energy-harvesting and -conversion devices specifically designed for building skins. These two skin systems share similar designs but utilize two different electrochemical technologies: RPEMFC and RFB. In the subsequent sections, the potential locations of the electrochemical devices are discussed. Furthermore, the module design complies with both the International Building Code (IBC) and the International Solar Energy Provisions (ISEP).

Four possible designs for utilizing the PV-RPEMFC(PV-RFB) are outlined and explained below. These designs include sunshade devices such as horizontal or vertical louvers, rainscreen cladding, spandrel glass, and double skins. Among the various options, rainscreen cladding and curtain wall systems stand out as the most feasible applications for the PV-RPEMFC(PV-RFB).

### 8.1. Components Needed and Hydrogen Storage Methods on Building Skins

The following essential elements are required to construct energy-harvesting, storage, and conversion panels within building skins:Photovoltaic panelsRPEMFCs or RFBsContainers for storing hydrogen or electrolyte solutionsElectrical wiring for connectionsGas and liquid (water or solution) conduitsAdditional components: insulation materials, air/water-tight fittings, and more.

Exploring various storage container locations, we present three schemes in Figure 1 to depict distinct energy storage locations as follows:Positioning a storage container between the photovoltaic panel and the RPEMFC (RFB). This configuration allows a single panel to handle all energy-related tasks: harvesting, conversion, and storage.An alternative plan involves providing each row of panels with its own dedicated storage container.The storage container could be situated in the building’s basement.

All the electrical wirings and conduits are hidden in PV-RPEMFC (or PV-RFB) panels as needed. The first option is favored within this research, and an assessment of these three methods will be conducted considering factors such as storage space, cost, and efficiency.

### 8.2. Shading System

Sunshades are exterior components installed on the building’s skins to provide solar control. They can be either fixed on the wall or designed to be movable. However, the weight of the sunshade system should be taken into consideration, as it may require its own structural supporting system if it becomes too heavy. The main challenges associated with sunshades are their susceptibility to wind load, their durability, and the need for regular maintenance. Building designs are continuously addressing these issues [47].

Implementing PV-RPEMFC(PV-RFB) systems in Figure 2 offers distinct advantages. These systems are independent of the building’s wall system, thereby eliminating concerns related to water/moisture infiltration or insulation problems. Moreover, if the PV-RPEMFC(RFB) system is designed to be movable, it becomes highly suitable for maximizing solar energy harvest due to its ability to trace the movement of the sun and adjust the panels accordingly.

### 8.3. Rainscreen

Another potential application for the designed PV-RPEMFC(RFB) is using it as cladding, specifically as a rainscreen cladding (shown in Figure 3), which serves as the protective outer layer of the building skins. Rainscreen cladding typically maintains a distance or gap from other exterior layers. They can be weather-resistant panels such as concrete, brick, or metal stud walls. These cladding panels are fixed to vertical fixing rails or brackets. The integration of PV modules into cladding systems is not uncommon, and since the outer layer of the PV-RPEMFC(RFB) is photovoltaic, it requires minimal modification to the overall exterior wall design. Additionally, the insulation layer remains separate from the cladding, minimizing potential thermal issues with alternative cladding components.

In the prototype design (illustrated in Figure 4), a size of 30 cm by 30 cm (1 foot by 1 foot) was chosen, as there are no specific size requirements outlined in the building codes. The design of the PV-PREMFC cladding panels takes into consideration the 2021 International Building Code and the 2021 International Solar Energy Provisions. As the outer skin of the building, the code specifies major considerations such as weather resistance, fire resistance, water resistance, and flood resistance.

To effectively function as the first line of environmental protection, the rainscreen panels must reduce rainwater penetration into the wall by minimizing holes and managing driving forces across the cladding [115]. To address this, the prototype utilizes a waterproof polycrystalline solar panel, which is resistant to water, acid, wear, and aging. The top of the prototype is capped with an aluminum sheet sloped to prevent water accumulation and facilitate drainage. This cap also protects the electrical components from water contact. At the bottom of the prototype, an extrusion is incorporated to prevent water from entering the gaps between panels. While these designs cannot entirely prevent rain, minor leakage into the cavity is normally removed by the natural airstream in the cavity.

Electrical safety measures are implemented by placing the wires in a box. Between the photovoltaic panels and reversible fuel cells at the bottom of the prototype, a channel or box is included to house air pipes, water pipes, and wires. The width of the channel will be modified based on the actual requirements. This channel is also electrically insulated and fire-resistant. This design not only ensures waterproofing but also takes aesthetic and fire safety considerations into account. Given the potential for freezing weather during winter, it is crucial to ensure a reliable water supply for the operation. To mitigate the risk of freezing, insulating the box or channel using thermal materials is recommended. This insulation will effectively minimize the possibility of water freezing and maintain a consistent flow.

The generated hydrogen will be stored in metal hydride hydrogen storage devices, where the hydrogen gas can undergo a reaction with metal instead of being compressed in containers. Related control valves will also be included as necessary.

During installation, the prototype will be securely fixed and screwed to studs for ease of installation. An air cavity with a minimum thickness of 3 cm is incorporated between the cladding and sheathing to allow for tubes and wires. Maintaining a proper distance between the PV panels and RPEMFC is also necessary to prevent efficiency reduction caused by excessive heat.

Given the presence of hydrogen pipes within the cladding system, the possibility of combustion must be addressed. The International Building Code, in section 1405, specifies limitations on the area, height, and fire separation for this type of wall system.

### 8.4. Integration with Curtain Wall for Use as Spandrel Glass

When a building is equipped with a curtain wall system, the spandrel areas present an excellent opportunity for integrating PV-RPEMFC (PV-RFB) technology. Spandrel glass is typically used to conceal floor panels or equipment. Since spandrel areas are susceptible to thermal bridging, which can lead to energy loss through building connections, it is important to think about insulation measures to reduce these effects. From an aesthetic standpoint, it is important for the spandrel areas to maintain consistency with the transparent vision glass. This ensures a visually cohesive appearance across the building facade. Integrating PV-RPEMFC (PV-RFB) technology within the spandrel areas of a curtain wall offers several advantages (illustrated in Figure 5) Firstly, it optimizes the use of space that would otherwise remain unused or serve solely as a visual cover. Additionally, it contributes to the generation of renewable energy while minimizing energy loss through thermal bridging. By carefully considering insulation and design aspects, the spandrel areas can seamlessly accommodate the PV-RPEMFC (PV-RFB) system, providing both functional and aesthetic benefits to the building.

### 8.5. Double-Skin (Ventilated Cavity)

Double-skin systems involve the use of an outer layer and an inner layer with a larger air cavity in between. This increased air cavity serves multiple functional purposes, including reducing sun exposure, enabling natural ventilation, providing an acoustic barrier, and creating space for maintenance [47].

In typical cases, the second skin or outer layer of a building is designed to be transparent or semi-transparent, allowing light to pass through. In this specific application, it is important for the photovoltaic (PV) elements to also be semi-transparent, ensuring that the light transmission is not obstructed. Additionally, the electrochemical components integrated into the building skin should not completely obstruct the view, preserving the visual aesthetics and maintaining transparency as much as possible. When integrating a double-skin system with a PV-RPEMFC (PV-RFB) system (illustrated in Figure 6), there are additional advantages to consider. For instance, the larger air cavity can facilitate cooling for the PV panels through the chimney effect, particularly when the surface of the panels becomes overheated. However, one crucial aspect to consider when implementing this integration method is the additional load it places on the building. Similar to sunshades, the question arises as to whether the double-skin system should have its own independent supporting structure or if the overall building design can accommodate the additional load.

Despite this consideration, implementing a double-skin system integrated with a PV-RPEMFC (PV-RFB) system is feasible and offers numerous benefits. These include reduced sun exposure, improved natural ventilation, enhanced acoustic performance, and the potential for cooling the PV panels. With proper design and structural support, this integrated approach can be successfully implemented.

## 9. Conclusions

The growing use of decentralized energy sources has led to a wide array of energy applications. However, the unpredictable nature of these sources presents challenges to the efficient and economical operation of traditional power plants. When these distributed sources are integrated into the main power grid, it can lead to mismatches between the supply and demand patterns, resulting in frequency problems, voltage problems, grid congestion, and power flowing in the wrong direction. As a result, there is a need for smaller-scale energy-storage solutions on the demand side to tackle these issues. Selecting the right energy-storage application for buildings that offer optimal and versatile use is a significant challenge, and factors such as cost, availability in the market, and policy considerations must be considered in decision making. To address this challenge, it is essential for researchers and designers to focus on creating cost-effective and user-friendly systems that encourage efficient and clean energy generation within buildings. In this context, electrochemical technologies have gained increasing importance, and expanding their applications has the potential to make them more commercially viable and affordable. Traditional energy storage in buildings primarily relies on batteries, valued for their cost-efficiency. While supercapacitors offer an alternative integrated into building skins, they suffer from limited scalability. In contrast, reversible fuel cells and flow batteries, apart from their storage capabilities, exhibit excellent scalability, allowing them to be tailored to the specific needs of a building. Given the constraints of space within building contexts, the integration of electrochemical devices into building skins presents a promising opportunity for energy harvesting, production, and storage. Technologies such as reversible proton exchange batteries and redox flow batteries, capable of operating within environmental temperatures, emerge as prime candidates for incorporation into building skin designs.

## Figures and Tables

**Figure 1 micromachines-14-02203-f001:**
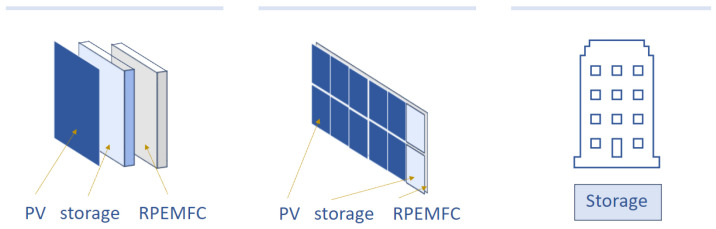
Possible storage locations.

**Figure 2 micromachines-14-02203-f002:**
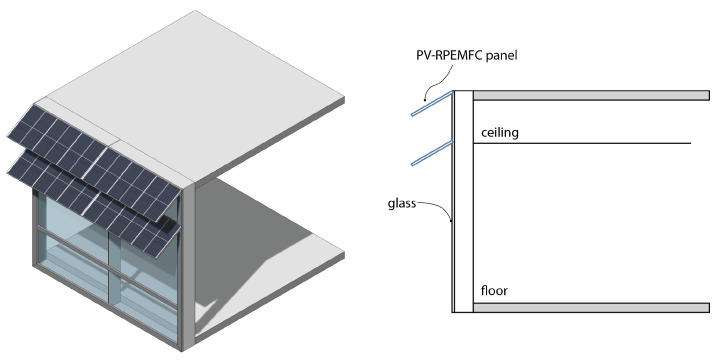
PV-RPEMFC used as sunshades on building skins.

**Figure 3 micromachines-14-02203-f003:**
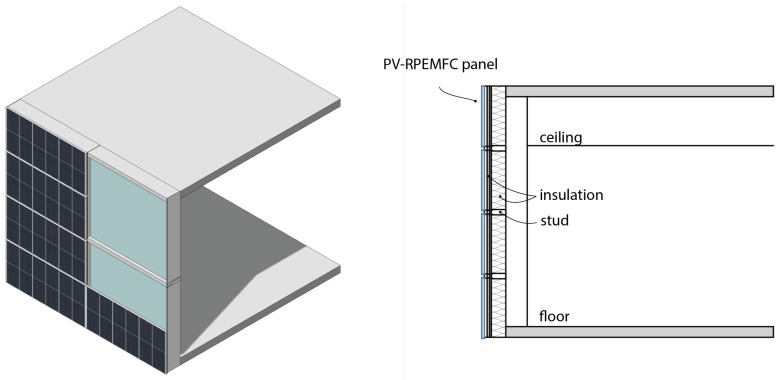
PV-RPEMFC used as raincreens on building skins.

**Figure 4 micromachines-14-02203-f004:**
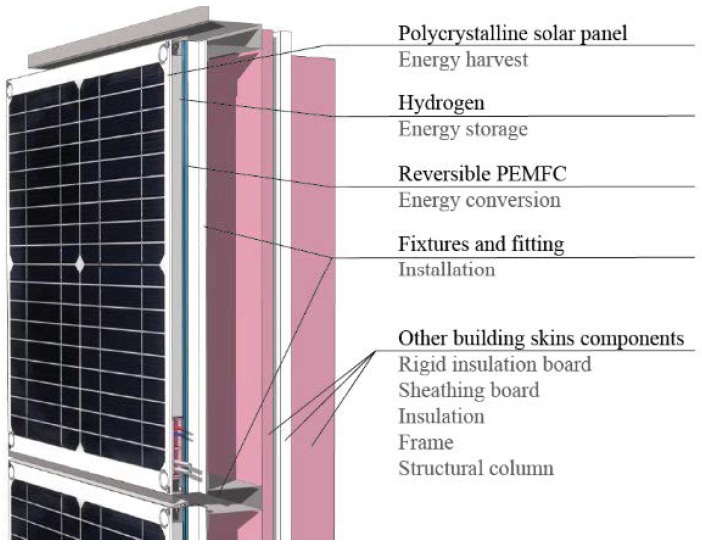
A tentative PV-PEMFC rainscreen panel design.

**Figure 5 micromachines-14-02203-f005:**
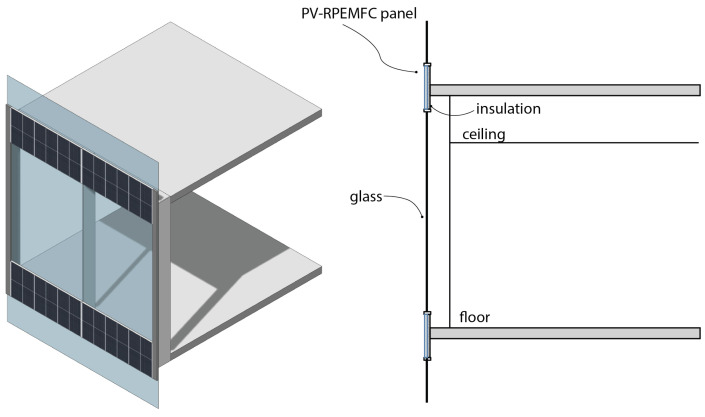
PV-RPEMFC used as spandrel on curtain walls.

**Figure 6 micromachines-14-02203-f006:**
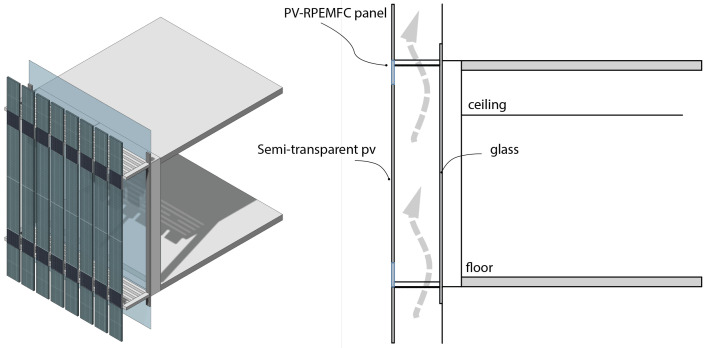
PV-RPEMFC used as the second skin.

## Data Availability

No new data were created or analyzed in this study. Data sharing is not applicable to this article.

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
