# Peer review of "A Review of Potential Electrochemical Applications in Buildings for Energy Capture and Storage"

_micromachines, 2023, doi:10.3390/mi14122203_

Round 1

Reviewer 1 Report

Comments and Suggestions for Authors

The authors have presented a review of potential electrochemical technologies and how they can be used to augment energy capture and storage efforts in buildings. Specific recommendations are listed below:                                                           

Specific Comments/Questions:                                                                

  1. The structure of the article is not particularly clear or intuitive – After section 1 introduces the topic, there is no clear flow for sections 2-5. Please consider revising and providing a clear logical flow for the article sections. Alternatively, sections 2-5 could be integrated into the "Introduction" as sub-sections.
  2. Introduction should also provide a stronger rational for the need for storing energy e.g., Authors state on line 218, page 5 “The electricity storage patches up the mismatch between load demand and renewable energy resources generation profile”. Please elaborate on the scope and magnitude of this mismatch – ideally using quantitative metrics and references. 
  3. Various technical terms such as energy density, power density have been used and appropriately explained in simpler terms – however the technical units should also be included (e.g., W/kg)
  4. Sections 6.1.1. and 6.2.1 discuss specific examples and case studies of battery/supercapacitor technology respectively. Why were these specific examples chosen? – Are they the current state of the art (SOA) in this technology?
  5. Furthermore, these examples need to be further analyzed ideally in tabular form such that important metrics such as Energy density (Wh/kg), Power density (W/kg), Specific capacity (Ah/g), C-rating, Average power, peak power, lifetime, and other metrics which can be easily compared by the reader – perhaps one of the core motivations behind reading a review article.
  6. Finally, any metrics discussed should be clarified if they belong to a device level architecture of pack-level architecture, as the two can differ widely even for the same underlying chemistry inside the energy storage device - this is especially relevant when considering commercial applications such as buildings.
  7. Similar considerations of device vs. pack-level apply to other technologies discussed herein e.g. line 344 page 7, “Although the energy conversion efficiency of this system tested at 2% is lower compared to the 17% efficiency achieved by a standard literature approach, it represents a new device capable of simultaneous energy harvesting and storage.”
  8. Page 11, lines 516, authors have chosen to discuss RFBs and R-PEMFC in more detail – please be more quantitative to describe and compare their features. Also, the acronym “RPEMFC” should be clearly defined at least once. Table 1 has some metrics – but these need to be expanded further with the current state of the art.  
  9. NFPA/ANSI/CSA standards have been quoted multiple times in section 7 – please provide references!

Reviewer 2 Report

Comments and Suggestions for Authors

This review emphasized the importance in adopting energy saving and energy storage technologies in building skins to improve the energy efficiency of buildings. The authors have demonstrated a well-architected structure of the manuscript and discussed several electrochemical energy storage technologies in details. The manuscript is suggested to be published after addressing a few comments.

1. The language throughout the manuscript is well-maintained. However, it is suggested to recognize the authors, for example, using XX et al. when discussing references instead of just giving the reference number. For example, line 300.

2. In the introduction part, the authors mentioned double-glazed panel in offering thermal comfort and noise-free environment for residents. As a typical energy-saving technology, electrochromic smart windows are suggested to be discussed, at least in the introduction part, if not as a part in section 6.

3. It would be great if the authors can tabulate the advantages and disadvantages of different electrochemical energy storage systems to be integrated with building skins. So that the readers can have a clearer comparison.

Reviewer 3 Report

Comments and Suggestions for Authors

See all my comments and questions in the attached pdf file.

BIPV technology must be explicitly mentioned in your text.

We need also more numbers / values for the different technologies that are proposed.

Comments on the Quality of English Language

My comments are also indicated in my attached pdf file.

Round 2

Reviewer 1 Report

Comments and Suggestions for Authors

In response to initial review, authors have mentioned that “These examples represent the most recent built projects, and some are sourced from the latest literature. The selection criteria are based on the following reasons: 1. Relevance to architecture or communities; 2. Recent research findings or real projects.”

This information should be explicitly included in the manuscript discussion so readers are aware of the selection criterion.

Other concerns have been sufficiently addressed.

Reviewer 3 Report

Comments and Suggestions for Authors

My comments and questions have been adequately considered and the needed updates introduced.